# Neuropsychological Assessments to Explore the Cognitive Impact of Cochlear Implants: A Scoping Review

**DOI:** 10.3390/jcm14217628

**Published:** 2025-10-27

**Authors:** Brenda Villarreal-Garza, María Amparo Callejón-Leblic

**Affiliations:** 1Otolaryngology Department, Virgen Macarena University Hospital, 41009 Seville, Spain; mcallejon@us.es; 2Biomedical Engineering Group, University of Seville, 41012 Seville, Spain

**Keywords:** cognitive decline, cochlear implantation, neuropsychological assessment

## Abstract

**Background/Objectives**: Hearing loss constitutes a modifiable risk factor for dementia. Auditory rehabilitation with devices such as cochlear implants (CIs) has been reported to prevent cognitive decline in older adults. However, post-implant cognitive effects remain highly heterogeneous across studies. Thus, the aim of this review is to synthesize the evidence on cognitive outcomes and their interplay with speech perception, quality of life (QoL), and psychological status. **Methods**: A bibliographic search was conducted following PRISMA guidelines from January 2015 to July 2025. Studies were eligible if they included adult CI candidates who completed cognitive and audiometric assessments. In total, 43 studies, including longitudinal and cross-sectional designs, were reviewed. Several studies also assessed hearing aid (HA) users and normal-hearing (NH) controls. Principal results were identified and analyzed across cognitive domains, audiological performance, QoL, and psychological outcomes. **Results**: CIs significantly improved cognition across longitudinal studies, with a higher number of assessments reporting gains in memory (61%), global cognition (57%), and executive function (46%); while attention, language, and visuospatial skills were less frequently evaluated. Though findings are not fully consistent, interactions between speech intelligibility and cognitive subdomains have also been found in several studies: global cognition (25%), executive function (22%), visuospatial skills (20%), attention (21%), language (17%), and memory (12%). Improvements in QoL, social engagement, depression, and anxiety are frequently observed. **Conclusions**: The lack of unified and adapted neurocognitive tools may prevent the observation of consistent outcomes across studies. Further research and multimodal data are still needed to fully understand the interaction between cognition, speech intelligibility, and QoL in CI users.

## 1. Introduction

Hearing loss (HL) is a prevalent sensory condition, affecting over 466 million people worldwide, including over 90 million adults [1]. In developed countries, nearly one in three adults aged 65–74 years, and half of those aged 75 years and older, experience disabling HL [2,3]. This health issue significantly impacts a patient’s quality of life (QoL), contributing to an increased risk of depression, loneliness, social isolation, cognitive decline, and lack of independence in daily activities [4,5,6,7]. Untreated HL might worsen the risk of dementia and mental health issues [8], thus highlighting the need for early interventions.

Recent studies explore the link between HL and dementia. The association was first highlighted by the Lancet Commission, which, based on meta-analyses, identified HL as a modifiable risk factor responsible for an estimated 7% of global dementia cases [9,10,11]. More recent investigations have focused on the effect of HL in different cognitive domains [12,13], including global cognition, memory, executive function, attention, language, visuospatial abilities, and reading skills [14].

However, the relationship between HL and cognitive decline remains an area of ongoing investigation, as a definitive causal link has yet to be established. Several hypotheses have been proposed to explain the potential mechanisms underlying the connection between the two conditions. The sensory deprivation hypothesis claims that reduced auditory input leads to cortical reorganization and brain atrophy, particularly in regions associated with auditory processing and higher-order cognitive functions, negatively affecting working memory, attention, and executive function [15,16,17]. The cognitive load hypothesis proposes that individuals with HL allocate a higher number of cognitive resources to decode degraded auditory signals, with fewer resources for higher-order cognitive tasks [18,19]. The common cause hypothesis adopts that HL and cognitive decline share common underlying risk factors, such as age-related neural degeneration, vascular pathology, inflammatory processes, or genetic factors, which contribute to the co-occurrence of these two conditions [15,17,19]. Lastly, the social isolation hypothesis highlights that HL often leads to social isolation and communication difficulties, which in turn increase the risk of cognitive decline [20]. Beyond its direct impact on communication, HL is strongly associated with reduced QoL and a high prevalence of psychological conditions such as depression, anxiety, and stress, thereby increasing vulnerability to social disengagement, loneliness, and reduced independence [21,22]. These are well-established risk factors for dementia and likely contribute to the underlying association between HL and cognitive decline [23].

Overall, these hypotheses suggest that hearing loss negatively impacts brain structure and function, increases cognitive effort, and limits social interactions, all of which impact cognition [24]. Researchers have been focused on investigating how interventions such as hearing aids (HAs) [25,26] and cochlear implants (CIs) offer a potential solution by restoring auditory input [14,27]. Mosnier et al. (2015) pioneered the research of CIs in older adults, showing a significant improvement in patients’ global cognition, episodic memory, processing speed, and executive function 12 months after implantation, especially in those with preoperative cognitive impairment [28]. A systematic review by An et al. [29], which included 20 research papers (648 subjects), concluded that rehabilitating hearing through CIs may have positive long-term effects on cognition, with different dynamics for different cognitive subdomains.

Cognitive functioning not only affects how individuals interpret and use auditory input but also their ability to adapt to HAs or CIs [30,31]. Auditory outcomes following cochlear implantation vary considerably among postlingual deaf adults. Although this variability is influenced by several factors, including age, preoperative hearing levels, duration of deafness, etiology, spectral and temporal resolution, etc. [32,33], these factors only explain part of the variability observed in large multicenter studies [34]. CI recipients often experience improved speech communication in quiet settings but continue to face challenges in noise [35]. Therefore, identifying the main neurocognitive domains related to better speech intelligibility has become increasingly important [36,37,38,39].

A meta-analysis by Amini et al. [40], based on 52 studies and 11 meta-analyses, showed that 50.8% of outcomes reported in the literature showed a positive association between CI and cognitive improvement, especially in memory, learning, and executive function domains. However, out of these studies, only 40.4% of outcomes reported an association between cognitive performance and speech intelligibility. These percentages clearly evidence the heterogeneity and sometimes contradicting outcomes reported in the literature thus far, possibly explained by the use of different audiological and cognitive tools, as well as varying study objectives. Despite the growing interest in the cognitive benefits of CIs and the wide range of available cognitive tests, there remains no consensus on the most suitable assessments or the specific cognitive domains to be evaluated. Therefore, characterizing the neuropsychological profile of CI patients remains an ongoing research issue that could provide valuable insight for multidisciplinary care approaches aimed at improving implantation outcomes and supporting personalized clinical decision-making [41,42,43,44].

Given the diverse and complex literature available, this study aims to provide a scoping review on the impact of CIs on cognition, measured through specific neuropsychological domains, and their interrelation with speech intelligibility and QoL. We qualitatively review and explore recent evidence on this topic and provide a set of recommendations that may help guide the design and implementation of future cognitive research and clinical approaches in CI users.

## 2. Materials and Methods

This study has been conducted following the Preferred Reporting Items for Systematic Reviews and Meta-Analyses (PRISMA) [45] from January 2015 to July 2025, see Figure 1. The PRISMA extension for Scoping Reviews-ScR [46] checklist can be found in Appendix A. No review protocol was registered for this study. The study selection process is illustrated in Figure 2.

### 2.1. Research Question and Added Value of This Study

The objective of this scoping review is to synthesize the recent evidence and provide a descriptive analysis of the influence of cochlear implantation on cognitive performance in adults, with special emphasis on specific measures reported over different cognitive domains and neuropsychological evaluations. To achieve this, the main questions/outcomes analyzed were: (i) Were there significant improvements in cognitive performance after CI? (ii) Were there significant differences in cognitive performance between CI users and other hearing profile groups? (iii) Did such improvements in cognition correlate with speech intelligibility and/or QoL outcomes?

Previous studies (e.g., [29,40]) have examined improvements in speech perception and cognitive outcomes following CIs, as well as the interrelationship between these domains. In the present work, we extend this analysis to additional health-related variables, including QoL, Hearing Related QoL (HRQoL), and psychological disorders such as isolation, depression, and anxiety. Furthermore, while prior reviews have primarily focused on longitudinal studies comparing pre- and postoperative outcomes, we also incorporate cross-sectional studies that evaluate differences across groups with different hearing profiles, e.g., HA users or normohearing (NH) individuals.

### 2.2. Search Strategy and Eligibility Criteria

The search strategy was designed using the PICOTs framework [47]. We considered studies published in English or Spanish with the following inclusion criteria:Participants: Adult patients aged 50 or older with postlingual severe-to-profound HL who were CI users or met the criteria for implantation.Intervention: Multi-electrode implant.Comparators: No constrictions are imposed. However, in the case of inter-group analysis, CI users were compared with unaided HL patients, HA users, and NH.Outcomes: gain in cognitive performance assessed through longitudinal neuropsychological evaluations conducted before and/or after cochlear implantation. Additionally, differences in cognitive performance across groups, and potential correlations between cognitive function, speech intelligibility, and QoL.Timeframe: Studies published in the last 10 years (2015–2025).

According to PRISMA recommendations, research was conducted in the following databases: PubMed, Scopus, Web of Science, Science Direct, Ebscohost, and APA PsycInfo. The search strategy was performed using medical subject headings (MeSH) terms: “(Hearing loss OR hearing impairment OR deafness) AND (dementia OR cognitive OR Alzheimer) AND cochlear implant”. In order to double-check the database search, we manually checked the reference lists of the studies included and performed both a backward and forward citation analysis. Following a criterion of novelty, thus ensuring the inclusion of contemporary CI technologies and evolving methodologies, the timeframe was limited to the previous 10 years (2015–2025). It is also important to note that this starting point coincides with the publication of a pioneering study by Mosnier et al. [28]), which laid the foundations for contemporary cognitive clinical research in CI users. Previous reviews have also started their analyses from the same foundational study [29,48].

The exclusion criteria were non-human studies, studies that do not involve at least one group of CI candidates or users, and those that did not provide a clear baseline cognitive assessment. Additionally, reviews, editorials, commentaries, and studies not directly relevant to the research question (e.g., those unrelated to cognitive evaluation) were excluded.

### 2.3. Study and Variable Extraction

Title and abstract screening were independently conducted by the two authors (B.V.-G. and M.A.C.-L.). Articles meeting the eligibility criteria were reviewed in full text. The same authors evaluated these full-text articles according to the predefined inclusion criteria. Any discrepancies were resolved through discussion and consensus.

Both authors (B.V.-G. and M.A.C.-L.) independently carried out the data extraction. The extracted variables included study design, inclusion/exclusion criteria, sample size, participant age, time of testing, cognitive assessments, audiometric evaluations, neuropsychological tests used, cognitive domains assessed, and additional variables.

### 2.4. Assessment of Study Quality and Risk of Bias

Most of the studies in this review used longitudinal and observational designs, in which participants were followed over time to track cognitive changes. Many of these studies were quasi-experimental, which means that there was no random assignment of participants to different groups. Researchers compared cognitive function before and after CI, sometimes using HA users or normal-hearing (NH) individuals as control groups.

Since these studies were not randomized, there is a high risk of bias, particularly in the selection of participants. Factors such as age, educational level, and duration of HL could have influenced the results. Some studies attempt to reduce these issues by matching participants or adjusting these factors in their analysis.

According to the Oxford Center for Evidence-Based Medicine (OCEBM) [49], quasi-experimental studies provide moderate-level evidence (Level 2 or 3), not as strong as randomized trials but still valuable for understanding everyday effects. To assess study quality, the Newcastle–Ottawa Quality Assessment Scale (NOS) for Cohort Studies [50] was used, which offers a practical and widely accepted approach to capture the key bias domains of selection, comparability, and outcome assessment in quasi-experimental designs.

### 2.5. Descriptive and Qualitative Analysis

A qualitative analysis was conducted by assessing the value and relevance of the studies included in this scoping review. Two authors (B.V.-G., M.A.C.-L.) independently reviewed the studies, and any discrepancies in the evaluation were resolved by consensus, with relevant observations reflected in the discussion section.

This review focused on monitoring individual cognitive changes and audiological outcomes, examining the differences in cognitive evaluations across studies, particularly when cognition was assessed before and/or after CI and/or through comparisons between different groups (e.g., CI users, CI candidates, HA users, and NH controls). Cognitive tests varied across studies, affecting how changes in cognitive function were interpreted. Some studies used a longitudinal design, evaluating participants before implantation and at follow-up intervals (e.g., 6, 12, 24 months or longer). Others relied on cross-sectional designs, comparing CI users to non-implanted hearing-impaired individuals or NH controls, which provided insight into group differences, but lacked baseline pre-implantation data. The cognitive assessments used across the studies varied in both the number of tests administered and the cognitive domains evaluated: while some studies only evaluated global cognitive screening, others conducted more comprehensive assessments. In total, a combination of 56 different cognitive tests/batteries was identified. In this review, we categorized these tests into seven subdomains: (i) global cognition, (ii) memory, (iii) executive function, (iv) attention, (v) language, (vi) visuospatial abilities, and (vii) reading skills (see Figure 2). These domains were chosen for their frequent assessment in the literature or defined in the Diagnostic Statistical Manual of Mental Disorders, Fifth Edition (DSM-5) [51,52]. Organizing the tests within this framework facilitates comparisons across studies and ensures consistency with common approaches in cognitive research. In addition, we summarized improvements in cognitive outcomes by noting whether each test showed a statistically significant change in longitudinal assessments or across groups (yes or no). This approach provides a qualitative overview across various studies and cognitive domains. We also listed whether such changes correlated with additional assessments such as speech intelligibility or other health-related variables, thus hopefully providing a broader understanding of the impact of cochlear implantation.
Figure 2Definition of cognitive subdomains assessed in this scoping review.
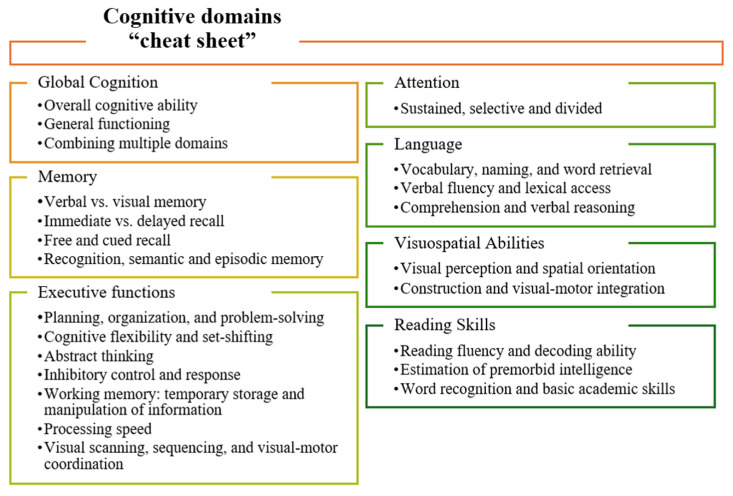


## 3. Results

The bibliographic search identified 7691 potentially relevant studies. After removing duplicates and applying the inclusion and exclusion criteria, 43 articles were included following the PRISMA flow diagram (see Figure 1). These studies incorporate a variety of designs, including prospective, longitudinal, observational, and cross-sectional studies. Included in this review, 34 studies employed a longitudinal design, with 26 studies exclusively evaluating cognitive changes over time for CI candidates. Moreover, five studies included an NH control group, two included HA users, and one study incorporated both NH and HA groups for comparative analysis. Additionally, nine studies used a cross-sectional design, and one used a mixed design. Of these, 5 assessed cognitive function across CI users, HA users, and/or NH controls, 3 focused exclusively on CI users, and two evaluated participants in the preoperative phase prior to implantation.

The sample sizes in the studies reviewed ranged from small cohorts of 7 participants [53] to larger groups of 103 participants [54]. The age of participants ranged from 20 years [54] to 80 years [55]. Ten studies incorporated participants who used HAs or had normal hearing, adding a comparative insight across groups. Inclusion criteria defined by the authors often consisted of postlingual onset of HL, corrected vision, and full proficiency in the language of the cognitive assessments. Most studies require CI candidates to have severe-to-profound bilateral sensorineural HL to comply with cochlear implantation criteria as required by their respective institutions.

Exclusion criteria included serious health conditions, neurological conditions, dementia or psychiatric disorders, and the inability to provide informed consent. Additionally, studies required participants to exhibit no signs of abnormal cognition, 15 of which required a minimum score on a screening test. Seven studies focused exclusively on cognitive screening tools to assess global cognition. Twelve studies employed cognitive screening measures, but also assessments targeting specific cognitive domains. The remaining 18 studies focused primarily on evaluating distinct cognitive domains such as memory, attention, and executive function.

The main findings of the reviewed studies are summarized in Table 1 and Table 2, organized by cognitive domain. Table 1 evaluates longitudinal studies assessing cognitive performance changes before and after cochlear implantation, highlighting significant improvements across domains. Additionally, it explores correlations between cognitive outcomes, speech intelligibility, QoL, and health-related variables. Table 2 reviews cross-sectional studies comparing cognitive differences across groups (CI users, HA users, and NH controls), reporting significant differences and correlations in audiological outcomes, QoL, and other health-related variables.

Appendix A compile the main characteristics and findings of the studies reviewed. Each table reports the following variables: reference of the study, design type, and quality rating according to the OCEBM, sample size, participant characteristics, time of testing, cognitive and auditory assessment, and other tests or variables included. In addition, the main cognitive outcomes and principal conclusions are reported. Appendix A focuses on longitudinal studies assessing cognitive function before and after cochlear implantation. Appendix A presents cross-sectional studies comparing cognitive performance between CI users and other control groups, such as NH individuals.

Additionally, Appendix A provides a detailed Quality Assessment of the cohort studies included in this review, using the Newcastle–Ottawa Quality Assessment Scale (NOA). This table evaluates the quality of cohort studies by assessing participant selection, comparability of study groups, and outcome assessment to determine the risk of bias and reliability of the reported results. All reviewed studies received seven or more stars out of nine, suggesting that the studies were generally of high quality (low risk). To further strengthen our assessment, we compared our NOS scores with those reported by Yeo et al. [35] for overlapping studies. Overall, our ratings were largely consistent, with most studies classified as low or moderate risk of bias. Minor differences in specific domains reflect slight variations in interpreting selection and comparability criteria, which is expected given the subjective nature of the NOS. Appendix A offers a description of the cognitive test employed across the reviewed studies, outlining the specific cognitive domains assessed, test objectives, scoring methods, and interpretation. This Appendix A facilitates the understanding of the cognitive measures used and supports the comparability of findings across studies.
jcm-14-07628-t001_Table 1Table 1Summary of Cognitive Test Outcomes, Speech Intelligibility, and Quality of Life Correlations After Cochlear Implantation.TestNWere There Significant Improvements After CI and Over Time?Did They Correlate with Speech Intelligibility Outcomes?Did They Correlate with QoL or Other Health Variables?ReferencesGlobal Cognition

AlaCog60Yes (6-m)-M3, 2-back, OSPAN, iFlanker, Memory recall (delayed)(12-m)Verbal fluencyOSPANNoNoVölter et al., 2018 [56]71Yes (12-m)-M3, delayed recall, 2-back, OSPAN, iFlanker, Verbal fluencyYes-WRS in quiet (M3, (β = −0.40)NoVölter et al., 2021 [57]71Yes (6-m)-OSPAN, iFlanker(12-m)-Memory recall, Verbal fluency(24-m)-2-backNoYes (baseline)-CRIq (OSPAN, τ = −0.33; verbal fluency, τ = −0.023)(12-m)-CRIq (TMT B, τ = −0.25)-GDS (verbal fluency, τ = 0.23)(24-m)-CRIq (M3, τ = 0.24)Völter et al., 2022 [58]75Yes (24-m)-Recall and Delayed recallNoNoVölter et al., 2023 [59]CANTAB23Yes (6-m)-simple RTI, SWM(12-m)-AST, PALbrief, SWM, strategy use, and simple RTIYes -WRS in quiet (SWM; *r* = −0.74)-SRS in noise (AST; *r* = −0.75)N/AJayakody et al., 2017 [60]CERAD 29NoNoN/AHuber et al., 2021 [61]CODEX18Yes (12-m)NoN/AAmbert-Dahan et al., 2017 [62]Cogstate Brief Battery 59Yes (18-m)-GMLTNoNoSarant et al., 2019 [63]101Yes (54-m)-GMLT-OBTNoNoSarant et al., 2024 [64]MoCA15Yes (12-m)Yes--Age at implantation (β = −0.33, *R*^2^ = 0.35);--PTA (β = −0.20, *R*^2^ = 0.28)--SRT in quiet (β = −0.008, *R*^2^ = 0.11)YesGDS (β = −1.07, *R*^2^ =0.68)Castiglione et al., 2016 [65]18Yes (12-m)NoN/AAmbert-Dahan et al., 2017 [62]77Yes (6-m)Yes-WRS in quiet (*r* = 0.314)N/AVasil et al., 2021 [66]MMSE93Yes (6-m)NoNoMosnier et al., 2015 [28]16NoNoN/ASonnet et al., 2017 [67]70Yes (7-y)NoN/AMosnier et al., 2018 [68]25Yes (6-m)Yes-WRS in noiseNoAnzivino et al., 2019 [69]44NoYes-WRS in quiet (*r* = 0.44)Yes-GBI physical health (*r* = 0.41).Sorrentino et al., 2020 [70]37NoNoNoGurgel et al., 2022 [55]53NoNoNoHerzog et al., 2022 [71]21Yes (1-y)N/AYes-NCIQ speech production (*r* = 0.47)Ohta et al., 2022 [72]21NoNoN/AZucca et al., 2022 [73]98NoN/ANoMosnier et al., 2024 [74]15NoNoN/ASchauwecker et al., 2024 [75]30NoNoN/AYoshida et al., 2025 [76]SAGE55NoN/AN/AYoung et al., 2023 [39]RBANS7Yes (3.7-y)-Coding, List Learning, Recognition and Recall, Story Memory and Recall, List Recall, and Semantic Fluency Yes (2-y)WRS in quiet (List learning: ↑6.65% [2.19–11.12])N/ACosetti et al., 2016 [53]20Yes (12-m)-Total score, Immediate memory, Attention, and Delayed memoryNoNoClaes et al., 2018 [77]24Yes (12-m)-Total score, Immediate memory, Attention, and Delayed memoryNoN/AMertens et al., 2021 [78]63Yes (12-m)-Total score-Immediate memory, Delayed memory-Visuospatial/constructional, Language, and Delayed memoryYes (pre-op) -SRS in noise (Immediate memory)(12-m)-WRS monosyllable in quiet (Visuospatial/constructional) and in noise-WRS disyllable in quiet (Visuospatial/constructional)-SRS in noise (Visuospatial/constructional)Yes (pre-op) -NCIQ Advanced sound *(r* = 0.43)-NCIQ Self-esteem (*r* = 0.48)-NCIQ Social functioning (*r* = 0.41)-HADS Anxiety (*r* = −0.50)-HADS Depression (*r* = 0.55)Calvino et al., 2022 [79]21Yes (12-m)-Total score, Immediate memory, Delayed memoryYesSRS in noise (*r* = −0.48)NoAndries et al., 2023 [80]25Yes (1-year)-Total score, Immediate memory, Attention, Delayed memory(4-year)-Immediate memoryNoN/AVandenbroeke et al., 2024 [81]WASI7Yes (3.7-year)-Matrix Reasoning, Block DesignYes (2-year and 3-year)WRS in quiet (WASI-VIQ: ↑12.20% [6.14–18.26], ↑12.43% [5.68–19.17]; WASI-FSIQ: ↑10.68% [4.03–17.34], ↑11.60% [4.70–18.50]; Vocabulary: ↑10.22% [4.75–15.69], ↑9.85% [3.40–16.29])N/ACosetti et al., 2016 [53]Memory5-Word Test93Yes (6-m)NoNoMosnier et al., 2015 [28]16NoNoN/ASonnet et al., 2017 [67]70NoNoN/AMosnier et al., 2018 [68]BVMT37Yes (12-m)NoNoGurgel et al., 2022 [55]Hopkins Verbal Learning Test–Revised (HVLT-R)37Yes (12-m)NoNoGurgel et al., 2022 [55]Rey Auditory Verbal Learning Test (RAVLT)25Yes (6-m)YesWRS in noiseNoAnzivino et al., 2019 [69]21NoNoN/AZucca et al., 2022 [73]Executive FunctionArithmetic21NoNoNoKnopke et al., 2021 [82]33Yes (2-year)NoNoHaeussler et al., 2023 [83]Digit Span25NoNoNoAnzivino et al., 2019 [69]37Yes (12-m)NoNoGurgel et al., 2022 [55]21NoNoNoKnopke et al., 2021 [82]21NoNoN/AZucca et al., 2022 [73]33NoNoNoHaeussler et al., 2023 [83]15NoYes (1-m)-SRS in quiet (*r* = 0.76) and in noise (*r* = 0.86)(3-m)-SRS in noise (*r* = 0.79)N/ASchauwecker et al., 2024 [75]Digit Symbol Coding21NoNoNoKnopke et al., 2021 [82]33NoNoNoHaeussler et al., 2023 [83]98Yes (6-m)NoNoMosnier et al., 2015 [28]Hayling Sentence Completion Test37NoNoNoGurgel et al., 2022 [55]Raven’s Progressive Matrices19NoNoN/AZhan et al., 2020 [84]Stroop Word Color Test29NoNoN/AHuber et al., 2021 [61]25Yes (12-m)NoNoAnzivino et al., 2019 [69]37Yes (12-m)NoNoGurgel et al., 2022 [55]19Yes (6-m)-Control and congruentYes -Incongruent-SRS in quiet (*r* = −0.48)N/AZhan et al., 2020 [84]15NoNoN/ASchauwecker et al., 2024 [75]Symbol Search21NoNoNoKnopke et al., 2021 [82]33Yes (12-m)NoNoHaeussler et al., 2023 [83]Trail Making Test A&B (TMT)93Yes (6-m)-B (12-m)-ANoNoMosnier et al., 2015 [28]7NoNoN/ACosetti et al., 2016 [53]16NoYes (6-m; TMT B)WRS in quiet (*r* = −0.72)N/ASonnet et al., 2017 [67]70Yes (7-year)NoNoMosnier et al., 2018 [68]25NoNoNoAnzivino et al., 2019 [69]37NoNoNoGurgel et al., 2022 [55]21NoYes (12-m)WRS in quiet (TMT A, *R*^2^ = 0.24, β = − 0.49)N/AZucca et al., 2022 [73]98NoNoNoMosnier et al., 2024 [74]Timed up and go (TUG)98NoNoNoMosnier et al., 2024 [74]Visual symbol Span19Yes (6-m)Yes-WRS in quiet (*r* = 0.60)-SRS in quiet (*r* = 0.50) and in noise (*r* = 0.49)N/AZhan et al., 2020 [84]Visual digit Span19NoNoN/AZhan et al., 2020 [84]Visual Object Span19NoNoN/AZhan et al., 2020 [84]RCPM30NoYesWRS in quiet (*r* = 0.367)N/AYoshida et al., 2025 [76]Attentiond2 Test of Attention93Yes (6-m) speed(12-m) number of errorsNoNoMosnier et al., 2015 [28]70Yes (7-year)NoN/AMosnier et al., 2018 [68]37YesNoNoGurgel et al., 2022 [55]Multiple Features Target Cancelation25NoNoNoAnzivino et al., 2019 [69]LanguageBoston Naming Test (BNT)7NoYes (2-year and 3-year)-WRS in quiet (↑10.23% [2.89–17.57], ↑5.38% [3.58–7.18])N/ACosetti et al., 2016 [53]Cardebat’s fluencies43Yes (3-m)NoN/ABaranger et al., 2023 [54]Controlled Oral Word Association Tests7NoNoN/ACosetti et al., 2016 [53]Test de Dénomination Orale d’images (DO80)16NoNoN/ASonnet et al., 2017 [67]Phonemic and semantic fluency tasks93NoYes -WRS in noiseNoMosnier et al., 2015 [28]70Yes (7-year)NoN/AMosnier et al., 2018 [68]25NoNoNoAnzivino et al., 2019 [69]21NoNoN/AZucca et al., 2022 [73]Visuospatial AbilitiesClock Drawing Test (CDT)93Yes (12-m)NoNoMosnier et al., 2015 [28]29Yes (12-m)Yes (3-m)-WRS in quiet (*r* = 0.55)-SRS in quiet (*r* = 0.52)N/AHuber et al., 2021 [61]70Yes (7-year)NoN/AMosnier et al., 2018 [68]21NoNoN/AZucca et al., 2022 [73]Corsi Block-tapping Test25NoNoNoAnzivino et al., 2019 [69]21NoNoN/AZucca et al., 2022 [73]Rey-Osterrieth Complex Figure Test16NoNoN/ASonnet et al., 2017 [67]25NoNoNoAnzivino et al., 2019 [69]Spatial span37NoNoNoGurgel et al., 2022 [55]Kohs block design30NoYesWRS in quiet (*r* = 0.538)N/AYoshida et al., 2025 [76]Reading skillsTest of Premorbid Functioning (TOPF)7NoYes (2-year)WRS in quiet (↑11.91% [3.83–19.99])N/ACosetti et al., 2016 [53]Test of Word Reading Efficiency (TOWRE)15NoNoN/ASchauwecker et al., 2024 [75]ReaCT Kyoto30NoNoN/AYoshida et al., 2025 [76]*r*: Spearman rank correlation, Pearson correlation, or partial correlation according to normality of distribution as reported in the studies; β: Normalized beta coefficient; τ: Kendall’s tau correlation coefficient; ↑##%: Multivariate logistic regression; N/A: Not Applicable.

## 4. Discussion

This scoping review examines the cognitive outcomes of cochlear implantation in older adults, with particular emphasis on psychological well-being and quality of life (QoL). Previous systematic reviews, such as those by An et al. (2023) [29] and Amini et al. (2023) [40], have primarily focused on auditory and neurocognitive outcomes, reporting improvements in global cognition and selected domains. However, these reviews do not assess the role of psychological well-being and QoL in determining long-term outcomes, as well as the influence of other contributing factors. The present review expands this scope by explicitly integrating assessments of psychological factors, QoL, and Hearing Related QoL (HRQoL), highlighting their interaction with cognitive function and auditory performance. Therefore, it offers a comprehensive perspective on the multidimensional impact of CI, highlighting that cognitive gains may be closely tied to improvements in emotional health, social engagement, and overall QoL, areas that remain underexplored in previous reviews.

Current evidence indicates that CIs exert a positive impact on multiple cognitive domains, particularly global cognition, executive function, memory, and attention, with additional improvements observed in language and visuospatial skills. However, the extent of these benefits varies considerably across studies, with some domains showing consistent gains and others remaining relatively stable over time. These findings highlight the complex and domain-specific nature of cognitive changes following CI and underscore the importance of evaluating both short-and long-term outcomes when assessing intervention benefits. For a detailed overview of the impact of CI on cognitive function, see Table 1.

### 4.1. Impact of Cochlear Implantation on Cognitive Function

#### 4.1.1. Global Cognition

Though findings across studies remain heterogeneous, improvements in global cognition after CI have been widely reported, especially those that used MoCA [62,65,66] and RBANS-H [53,77,78,80]. The MMSE demonstrated the least impact of change [55,67,70], with only one study [72] reporting improvements one year post-implantation. Generally, improvements in global cognition have been more evident in participants with preoperative cognitive impairment (MMSE ≤ 24), thus supporting the idea that CI may reduce sensory deprivation effects and promote cognitive recovery.

#### 4.1.2. Memory

Memory outcomes following CI are generally associated with improvements in episodic memory, immediate and delayed recall one year post-implantation [53,55,60,69,77,79,80]. Evaluation of these functions using the RBANS and ALAcog batteries showed some of the most consistent post-implantation changes, as evidenced by the longitudinal studies by Völter et al. [56,57,58,59], which consistently documented enhancements in memory following 12 months of CI use. Sarant et al. [64] also reported stable delayed recall and visual memory after 4.5 years. Together, these findings suggest that restoring auditory input through CI may help preserve memory by reducing cognitive load, thereby supporting neural mechanisms involved in memory recovery [85]. However, other studies, such as that of Vandenbroeke et al. [81], noted a decline over four years. Huber et al. [61] reported non-significant improvement in both verbal and figural episodic memory one year post-implantation. These mixed results show that the effects of CI on memory remain unclear, with longer-term studies being still needed to better understand these outcomes.

#### 4.1.3. Executive Function

Executive functions represent the most repeatedly assessed domain in the literature, demonstrating variable improvements following CI, with 47% of the studies reporting gains. As illustrated in Appendix A, this domain accounts for the largest number of evaluations and the greatest combined sample size across studies, thereby providing greater robustness to the findings. When examining carefully specific tests evaluating executive functions that led to significant improvements following implantation, the N-back, OSPAN, and Flanker task from the ALA cog battery [56,57,58], Stroop Word Color Test [55,69,84], and Cogstate Brief Battery (Groton Maze and One-back) [63,64] exhibited consistent improvements over different follow-up periods.

These findings provide further support to the cognitive load hypothesis, which posits that alleviating sensory deprivation frees cognitive resources that can then be allocated to higher-order processing [17]. The improvements in cognitive flexibility and inhibitory control reported by Sonnet et al. [67] and Völter et al. [58] further underscore the contribution of neuroplastic mechanisms following sensory restoration [86,87]. In contrast, Huber et al. [61] and Mosnier et al. [68] reported minimal or non-significant changes, highlighting the substantial inter-individual variability observed in cognitive outcomes.

Consistent gains in processing speed were observed in most studies one year post-implantation [53,60,67,82,84], with performance remaining stable after two years [82,83]. Working memory also demonstrated improvement across multiple studies, one and two years after implantation [56,57,67,82,83,84], with evidence of sustained enhancement up to 4.5 years post-surgery [64]. Conversely, nonverbal reasoning, as assessed by Zhan et al. [84] using Raven’s Progressive Matrices, did not show significant post-implantation changes, suggesting the need for further exploration in this domain.

#### 4.1.4. Attention

Although attention scores exhibited considerable gains in several studies [56,57,58,77,78,81], varying degrees of improvement and stability were observed; while some reported an enhancement in simple (spatial span), and sustained attention (d2 test) [55], others [68] detected a decline one year post-CI; and only minor improvements have been observed in the Trail Making Test [58,67,69]. Stable psychomotor function (Detection test) was found in [64] after 18 months, with minimal changes observed over a 4.5-year follow-up period [63]. These findings may suggest that cochlear implantation enhances specific domains of attention, particularly selective, sustained, and focused attention. However, long-term outcomes vary, as attention often declines with age, which may affect speech comprehension in older adults [88].

#### 4.1.5. Language

In terms of language, phonemic and semantic fluency improved following cochlear implantation in various studies [53,54,57,58,78,79]. Sonnet et al. [67] showed stable language performance over time, with no significant changes post-implantation using the DO-80 test. These improvements may demonstrate the role of CIs in enhancing cognitive function by supporting language recovery, especially verbal fluency. However, outcomes varied across studies, potentially due to the different tests used.

#### 4.1.6. Visuospatial Abilities

Visuospatial abilities are typically assessed through visually based cognitive tasks. Findings on visuospatial skills post-implantation varied, with some studies showing notable improvements [55,60] and others showing a significant decline over four years [81].

The older cohort (≥60 years) in [79] demonstrated significant improvement in the visuospatial/constructional domain of the RBANS-H at 12 months post-implantation. These findings suggest that auditory input from a CI may contribute to cognitive enhancement by activating brain regions beyond the auditory cortex. This aligns with previous research, which demonstrated that auditory stimulation can interact with non-auditory neural networks, including posterior and parietal regions that support visuospatial processing [89,90].

#### 4.1.7. Reading Skills

In the studies reviewed, reading skills were primarily assessed as part of the inclusion criteria to ensure that participants could comprehend instructions during cognitive assessments [60,84]. In some cases, reading skills were also included within the cognitive test battery [53,75,76]. However, no significant improvements or changes were observed.

### 4.2. Cognitive Differences Across Profile Groups

While CI users may not achieve the same cognitive performance as their NH peers, they often outperform HA users, which highlights the potential cognitive benefits of cochlear implantation beyond auditory restoration. These cognitive differences were observed in most of the cognitive domains studied, as explained below and outlined in Table 2 and Appendix A.
jcm-14-07628-t002_Table 2Table 2Summary of Cognitive Tests: Group Differences, Correlations with Speech Intelligibility, and Quality of Life Outcomes.TestNWere There Significant Differences in Cognitive Performance Across Groups?Did They Correlate with Speech Intelligibility Outcomes?Did They Correlate with QoL or Other Health Variables?ReferencesGlobal Cognition

MoCA45YesNoNoCastiglione et al., 2016 [65]NIH Toolbox Cognition Battery20NoYes-Pattern Comparison Processing Speed Test (*R*^2^ = 0.28, β = −0.23)N/ASchvartz-Leyzac et al., 2023 [91]MMSE31N/ANoNoMoberly et al., 2018 [30]RBANS142Yes-Total scoreYes-WRS in quiet (CI: *r* = 0.313, NH: *r* = −0.256)-SRT in noise (CI: *r* = −0.354, NH: *r* = −0.354)Education (CI: *r* = 0.332)Claes et al., 2018 [92]30N/AYes (Attention)-PTA (250 Hz: *r* = −0.58, 500 Hz: *r* = −0.40, 2000 Hz: *r* = −0.45, 4000 Hz: *r* = −0.47)-WRS in noise (SNR10: *r* = 0.66, SNR+5: *r* = 0.58)-SRS in noise (SNR5: *r* = 0.40) NoGiallini et al., 2023 [93]MemoryCVLT-II102YesNoNoKramer et. al., 2018 [94]Non-Verbal Learning Test (NVLT)61NoNoYes-APHAB Ease of Communication (*r* = 0.50)Huber et al., 2023 [95]Executive FunctionAuditory Stroop Task93YesNoN/ACeuleers et al., 2024 [96]Categorization Working Memory Task (CWMT)30N/AYes -PTA (250 Hz: *r* = −0.50, 500 Hz: *r* = −0.41)-WRS in noise (SNR+10: *r* = 0.50, SNR+5: *r* = 0.52)NoGiallini et al., 2023 [93]Digit Span31N/AYes-WRS in quiet (*r* = 0.38)-SRS in noise (*r* = 0.78)NoMoberly et al., 2018 [30]30N/AYes (Forward)-PTA (250 Hz: *r* = −0.40)-WRS in noise (SNR+10: *r* = 0.40)(Backwards)-PTA (1000 Hz: *r* = −0.41, 4000 Hz: *r* = −0.54)NoGiallini et al., 2023 [93]Go/No-Go Test61NoNoYes -APHAB Aversive (*r* = 0.35)Huber et al., 2023 [95]Letter-Number Sequencing Task93YesNoN/ACeuleers et al., 2024 [96]N-Back61NoNoNoHuber et al., 2023 [95]Raven’s Progressive Matrices102YesNoYes-Socioeconomic status (*r* = 0.19)Kramer et. al., 2018 [94]31N/AYes-WRS in quiet (*r* = 0.38)NoMoberly et al., 2018 [30]97YesYes-SRS in quiet (IEEE, NH: β = 8.2 [1.4–15])N/AMoberly et al., 2025 [97]Stroop Word Color Test102NoNoNoKramer et. al., 2018 [94]31N/ANoNoMoberly et al., 2018 [30]97NoYes-Audiovisual SRS (NH, β = 14.6 [4.3–24.9])N/AMoberly et al., 2025 [97]Trail Making Test A&B (TMT)17N/AYes-WRS in quiet (CI-only: *r* = −0.52, bimodal: *r* = −0.75) -SRS in noise (*r* = 0.55) NoHua et al., 2017 [98]61NoNoYes (B)-APHAB Ease of Communication (*r* = 0.32)Huber et al., 2023 [95]Visual symbol Span102YesNoNoKramer et. al., 2018 [94]Visual digit Span102NoNoYesSocioeconomic status (*r* = 0.23)Kramer et. al., 2018 [94]97YesNoN/AMoberly et al., 2025 [97]Visual Object Span102NoNoNoKramer et. al., 2018 [94]Reading Span Test17N/AYes-WRS in quiet (bimodal, *r* = 0.71)N/AHua et al., 2017 [98]AttentionLetter Detection Test93YesNoN/ACeuleers et al., 2024 [96]LanguageRegensburg Word Test (RWT)61NoNoYes (phonemic)-APHAB Ease of Communication (*r* = 0.34)(Semantic)-APHAB Aversive (*r* = 0.35)Huber et al., 2023 [95]WordFAM97YesYes-Audiovisual SRS (NH, β = −1 [−1.4–0.6])N/AMoberly et al., 2025 [97]Reading skillsTest of Word Reading Efficiency (TOWRE)102YesNoYes(Non-words and words)-Socioeconomic status (*r* = 0.26 and *r* = 0.37)Kramer et. al., 2018 [94]31N/ANoNoMoberly et al., 2018 [30]97YesYesCI users-WRS in quiet (β = 2.9 [0.9–5])-SRS in quiet (IEEE, β = 14.6 [4.3–24.9])-SRS in quiet (PRESTO, β = 14.9 [2.4–27.4])N/AMoberly et al., 2025 [97]Wide Range Achievement Test (WRAT)31N/ANoNoMoberly et al., 2018 [30]*r*: Spearman rank correlation, Pearson correlation, or partial correlation; β: Normalized beta coefficient; N/A: Not Applicable.

#### 4.2.1. Global Cognition

Cognitive performance varies across different hearing profiles, specifically among CI users, HA users, individuals with untreated hearing loss, and NH peers. NH individuals consistently outperform CI users, who do not fully reach comparable cognitive levels as measured by the RBANS-Total [78,92]. However, compared to untreated patients, CI users show less cognitive decline [64], alongside significant improvements in MMSE scores observed in both CI and HA groups, which then stabilized over a period of 12 months [69]. CI recipients also achieved post-implant MoCA scores comparable to long-term HA users and NH controls [65,70], thus suggesting that global cognition in CIs can compete with NH levels, especially when considering age, education, and controlled testing conditions.

#### 4.2.2. Memory

CI users frequently outperform unaided hearing-impaired individuals [59]. While CI users match NH controls in figural episodic memory (immediate recall), deficits persist in verbal episodic memory and delayed recall [61]. NH individuals consistently outperformed CI and HA users on immediate working memory tasks (letter-number sequencing) [96] and verbal learning (California Verbal Learning Test II) [94].

#### 4.2.3. Executive Function

Executive functions, particularly cognitive flexibility and inhibitory control, differ across hearing groups. NH individuals consistently outperform HA and CI users on tasks such as the Auditory Stroop [96]. Although CI users show notable post-implantation gains [28,64], persistent deficits remain—especially among older recipients—likely reflecting the combined influence of auditory and age-related cognitive decline [61].

CI users also tend to exhibit slower reaction times than NH controls [60,82,91], a pattern often attributed to the additional cognitive effort required to process degraded auditory input through electrical stimulation. Psychomotor performance, however, appears to remain stable up to 4.5 years post-implantation, whereas NH controls show significant age-related declines over the same period [64].

Consistent improvements have also been reported in nonverbal reasoning and working memory tasks [97]—including Raven’s Progressive Matrices, Symbol Span, and Visual Digit Span—with experienced CI users outperforming CI candidates [94].

#### 4.2.4. Attention

NH participants outperformed both HA users and CI recipients on measures of selective attention, such as the Trail Making Test A and the Stroop Test [61]. However, over a 4.5-year follow-up, Sarant et al. [64] reported that CI recipients maintained stable attention levels, whereas NH controls showed a decline. This suggests that hearing restoration may not only improve baseline attention deficits but also help mitigate cognitive decline over time.

#### 4.2.5. Language

Language appears to be variably affected in CI users compared with NH individuals. Claes et al. [92] and Huber et al. [61] found no significant differences between groups on the RBANS and the Regensburg Word Test (RWT). In contrast, Moberly et al. [97] reported lower vocabulary scores (WordFAM) in CI users with more than one year of experience.

#### 4.2.6. Reading Skills

Even after controlling for socioeconomic status and reading proficiency, NH peers still outperformed both experienced CI users and candidates in lexical access speed, as measured by the TOWRE [94]. In contrast, other studies did not find any significant differences.

### 4.3. Speech Perception Outcomes and Cognitive Influences

Speech perception outcomes, particularly in quiet listening conditions, show consistent improvement in CI users, reflecting the overall effectiveness of the intervention [63,64,70,73]. Although the remaining difficulties faced by CI users with speech in noise [71,75,84,96] have been partly attributed to cognitive factors—notably reduced working memory capacity, slower processing speed, and diminished selective attention—which may hinder the ability to separate speech from background noise [99,100]; improvements in speech perception have not always been consistently correlated with changes in neurocognitive performance, see Appendix A. Therefore, this suggests that their trajectories might progress, to some extent, independently, and that further research is needed to clarify the interplay between auditory restoration and cognitive function.

#### 4.3.1. Global Cognition

Neurocognitive factors have been shown to influence speech recognition in some cases [65,66,69,77], and may account for part of the variability observed in speech in noise performance [69,80,92]. These findings call for further research to better understand the influence of cognition on CI adaptation and underline the bidirectional relationship between auditory and cognitive processing [98,101].

#### 4.3.2. Memory

Memory appears to have only a weak association with speech perception outcomes: while correlations with sentence recognition in noise have been observed using the Rey Auditory Verbal Learning Test [69] and the RBANS immediate and delayed memory indices [53]; no significant associations have been reported for other memory measures, including the 5-Word Test [67,68], Brief Visuospatial Memory Test (BVMT) [55], and Hopkins Verbal Learning Test–Revised (HVLT-R) [55].

#### 4.3.3. Executive Function

Executive function represents one of the most frequently studied domains in the exploration of the relationship between cognition and speech intelligibility. Specifically, 22% of the studies reported a significant correlation, suggesting its role as one of the most consistent predictors of speech performance. Components such as working memory (Digit and Visual Symbol Span) [75,84], cognitive flexibility and processing speed (TMTA&B) [58,67,73] may emerge as potential determinants of speech perception, particularly under challenging listening conditions [44,75]. This aligns with the ELU model, which highlights the need for strong executive resources to resolve degraded speech input. Rather than being a consequence of reduced listening effort, this complex domain appears to play a central role in supporting successful CI speech comprehension [102].

#### 4.3.4. Attention

Attention has been shown to play a role in speech-in-noise processing, with the Attention Index [93] and the CANTAB Attention Switching Task [60] significantly correlated with better sentence recognition in noise. However, its relation under quiet listening conditions seems to be more limited, as weaker or no associations have been found in this case [30,57,91].

#### 4.3.5. Language

Language abilities appear to play an important role in predicting speech recognition outcomes, with sentence recognition indirectly influenced by auditory–verbal integration and vocabulary knowledge [53,97]. Moreover, phonemic and semantic fluency have been associated with improved word recognition in noise and improved long-term speech perception outcomes in CI users [28].

#### 4.3.6. Visuospatial Abilities

The contribution of visuospatial abilities to speech perception may be limited or test-dependent. Although they appear to influence speech perception outcomes, as indicated by the Clock Drawing Test [61], Kohs block design [76], CANTAB Spatial Working Memory [28], and the RBANS visuospatial index [79], other tests found no significant associations [28,69,73,74].

#### 4.3.7. Reading Skills

Reading skills have been associated with word recognition in quiet, as measured by the Test of Word Reading Efficiency [53] and the Test of Premorbid Functioning [97].

### 4.4. Factors Influencing Cognitive and Auditory Outcomes

#### 4.4.1. Age

Age is a complex factor influencing cognitive and auditory outcomes in CI users, as both naturally decline with healthy aging. While some studies have found minimal associations between age and cognitive or auditory outcomes [75], others have negatively impacted outcomes in CI users, reflecting lower cognitive improvement [79], slower processing speeds [91], and poorer sentence-in-noise recognition [73]. Age also influences memory performance, with younger females achieving higher scores in the RBHANS- immediate recall [59]. The reported effects of age on cognitive and auditory performance may not be solely and/or directly related to hearing loss, as executive function tends to decline in normal aging, thus highlighting the need to control for age when interpreting outcomes in older CI patients.

#### 4.4.2. Education

Education strongly influences cognitive performance in CI users [83], possibly because higher education provides greater cognitive reserve, which can help compensate for auditory deprivation and support adaptation to the implant [58,61]. Higher education was linked to greater performance in attention [93], memory [59], and global cognition [92]. Conversely, individuals with lower baseline cognitive reserve led to greater postoperative improvement, thus suggesting ceiling effects in highly educated groups [58,66,84]. These findings align with the cognitive reserve hypothesis [103,104], thus highlighting the need to consider educational background in the interpretation of cognitive assessments.

#### 4.4.3. Rehabilitation

Rehabilitation is essential for optimizing post-CI cognitive and auditory outcomes [80]. Ambert-Dahan et al. [62] reported that all participants who underwent auditory-cognitive rehabilitation had improved verbal processing, attention, and mental flexibility.

#### 4.4.4. Other Influencing Factors

The impact of other demographic and clinical factors on cognitive outcomes post-implantation has also been explored. Duration of hearing loss, gender, and etiology were found to have minimal effects on CI outcomes [73,75], whereas differences in socioeconomic status, vocabulary knowledge, and gender partially explained group differences in cognitive performance [94]. When these factors were statistically controlled, group differences in several cognitive tasks were reduced or became no longer significant, emphasizing their effects on auditory-cognitive outcome.

### 4.5. Long-Term Cognitive Stability and Decline

Longitudinal studies included in this review indicate that cognitive benefits following CI generally persist over time, although delayed memory may decline gradually [68,81]. The stability in global cognition has been observed over follow-ups of four to seven years [58,64,68,71,76,81], with only minimal progression to dementia among participants with pre-implant mild cognitive impairment [68]. Nevertheless, long-term cognitive trajectories require further investigation, as age-related neurodegeneration may offset initial improvements after CI [105,106].

### 4.6. The Impact of Cochlear Implants on Mood Disorders and Quality of Life

Beyond gains in cognitive function, CI has been associated with significant enhancements in QoL, including reduced social isolation and improved emotional well-being [63,79]. Studies using the Nijmegen Cochlear Implant Questionnaire (NCIQ) and the Glasgow Benefit Inventory (GBI) confirmed post-implantation gains in social engagement and overall QoL [70,81,92], thus highlighting the broader psychosocial benefits of CI and their role as therapeutic tools beyond auditory rehabilitation [107]. HRQoL studies have examined patients’ subjective hearing improvements, which have been linked to better self-perceived sound quality and hearing, as measured by HISQUI19 and SSQ12 [78].

Reduced levels of anxiety, depression, and social isolation have also been observed. Improvements in anxiety and depression, along with enhanced self-esteem and social functioning, have been reported following CI intervention [65,77,79]. CI recipients also showed decreased negative affectivity, lower Type D personality trait scores, and reduced stress [60,78,81], reflecting positive emotional and personality changes. These findings align with the social isolation hypothesis, which proposes that hearing loss leads to social isolation and emotional challenges, eventually accelerating cognitive decline. By enhancing auditory input, CI supports both emotional well-being and cognitive function [108,109].

However, not all studies reported psychological benefits. Some reported no significant changes in stress, depression, or anxiety [67,82,83], indicating that CI effects on long-term psychological well-being vary among patients [60,72,79].

The association between cognitive performance in CI users and QoL has been examined to understand the influence of cognition on psychosocial outcomes and overall well-being following CI, see Appendix A. Interestingly, self-reported subjective hearing, as measured by the APHAB, correlated with improvements in cognitive flexibility, phonemic fluency, and visual episodic memory [95]. Cognitive Reserve Index Questionnaire scores have been associated with executive functions, attention, and verbal fluency [58], while verbal fluency was also linked to global cognitive screening (MoCA) [65]. MMSE performance correlated with physical health on the GBI [70] and with NCIQ speech production [72]. Additionally, RBANS total scores were associated with NCIQ outcomes and HADS anxiety and depression [79]. To fully understand these correlations and clarify whether these relationships reflect true associations or causal effects, a multimodal approach is necessary, integrating cognition, health-related factors, and their interaction with auditory rehabilitation.

### 4.7. Limitations of This Study

Several methodological limitations should be considered when interpreting the results of our review. A key limitation concerns the substantial variability in cognitive assessment tools used across studies, which hampers the comparability of outcomes. In the absence of a standardized cognitive framework, we adopted the DSM-5 framework; however, most tests inherently involve multiple cognitive processes, complicating their assignment to discrete domains. Consequently, the percentages and figures reported in this manuscript are contingent upon this classification framework, and the use of an alternative taxonomy could yield different distributions and numerical outcomes.

In addition, our methodology treats all cognitive tests equally, which may oversimplify the true effects of cochlear implantation on cognition. Tests showing minimal or barely meaningful improvements are weighted the same as those demonstrating substantial changes, and variations in sample size and test characteristics are not accounted for. The considerable heterogeneity in study designs, outcome measures, and statistical reporting demands a meaningful quantitative synthesis, which is beyond the scope of this scoping review. Although we cannot offer evidence on which battery is the most appropriate to use based on the available data, we do provide a series of practical recommendations and highlight existing clinical limitations in the current literature that may guide other researchers or clinicians at the outset of their studies. Future research employing standardized effect sizes or meta-analytic techniques could provide a more precise and detailed understanding of cognitive outcomes in this population. Finally, no review protocol was registered for this study, representing a methodological limitation due to the potential risk of selective reporting bias. To mitigate this concern, we adhered strictly to PRISMA 2020 guidelines, and all outcomes of interest are comprehensively reported to ensure transparency and reproducibility.

### 4.8. Future Challenges and Directions

As research on cognition in CI users advances, several challenges remain to be addressed. Drawing on insights from this scoping review, we outline future directions that may help improve the consistency and validity of evidence in this field.

#### 4.8.1. Interplay Between Hearing Loss and Dementia

HL and dementia appear to share several common risk factors, including poor nutrition, visual impairments, cardiovascular issues, frailty, poor physical health, depression, and social isolation [4,110]. Although this relationship often weakens after accounting for age-related factors [111], vascular dysfunction appears to underlie both HL and cognitive decline. [112]. Age-related HL is commonly observed in patients with dementia, and it is associated with lower cognitive performance and QoL [26]. This suggests that sensory decline, particularly in hearing, may serve as an early marker for identifying individuals at risk of accelerated cognitive aging. An indirect pathway may exist, with HL increasing the likelihood of social withdrawal, which in turn increases the risk of dementia [113]. Given the heterogeneity of both dementia and HL, examining their subtypes may provide valuable insights into their complex relationship and support the development of more targeted and effective interventions [26]. Ultimately, while cochlear implantation appears to mitigate cognitive decline, its potential role in preventing dementia remains unclear. Large-scale, randomized controlled trials are necessary to establish causal relationships between CI and a reduced risk of dementia.

#### 4.8.2. Definition of Normal Cognition

In studies assessing cognition in CI candidates or users, individuals with abnormal cognition, cognitive impairment, or a diagnosis of dementia are frequently excluded. Across the reviewed studies, three main approaches have been identified to assess and classify cognitive status. The first approach applies cognitive screening cut-off scores to distinguish "normal" from "abnormal" cognition (e.g., MoCA ≥ 26 or MMSE ≥ 24), sometimes supplemented with additional tests such as the WRAT, requiring a total score ≥ 70 [84]. The second approach involves the application of variable cut-off scores depending on contextual factors such as normative data, age, and education level (e.g., MMSE ≥ 26 [94], MoCA ≥ 23 for those aged ≥ 60 [96], WRAT ≥ 75 [94], or RBANS-H total score above the 16th percentile [80]). The third approach is the exclusion of participants based on a prior clinical diagnosis of cognitive impairment or dementia. This variability evidences the absence of standardized criteria for evaluating cognitive status in CI research: while excluding individuals with significant impairment may reduce confounding, it also limits the generalizability of findings. This is especially relevant as cognitive impairment is increasingly common among older adults, who represent a growing proportion of CI candidates.

#### 4.8.3. Need for Standardized Cognitive Tests

Despite efforts to ensure participant comprehension of test instructions, variability in test administration and adaptation remains a major limitation. This reflects the lack of standardized cognitive tests specifically designed and validated for the CI population, with 56 different tests and variable cut-off scores employed. This heterogeneity introduces a risk of systematic assessment bias and potential overdiagnosis, consistent with the Harbinger Hypothesis, whereby inconsistencies in test selection and administration may lead to misleading indications of cognitive decline and/or to an overestimation of cognitive deficits. Such variability reduces the reliability and comparability of findings across studies. Many cognitive tasks heavily rely on verbal instructions or auditory processing, placing hearing-impaired individuals at a disadvantage [19,114]. Addressing this issue underscores the clinical importance of using auditory-adapted cognitive tests, which minimize the confounding effects of hearing loss and allow for a more precise evaluation of cognitive functions in CI recipients.

#### 4.8.4. Statistical Reporting Variability

Variability in the statistical methods and parameters reported can affect the interpretation of results. Here, the importance of including effect sizes, *p*-values, confidence intervals, correlation coefficients, and regression parameters is essential. Providing open-access datasets could enhance transparency and reproducibility. Integrating these datasets within the emerging field of computational audiology [115] could enable the development of advanced predictive models, support early identification of individuals at risk, and develop personalized auditory-cognitive rehabilitation strategies [116,117].

#### 4.8.5. Neurobiological and Computational Perspectives

Reduced hearing ability has been associated with decreased brain volume in both lateral and medial temporal regions, with sex-related differences observed [118,119]. Additionally, higher genetic risk for HL has been associated with poorer cognition and reduced gray and white matter integrity, while elevated cerebrospinal fluid (CSF) tau levels and brain atrophy appear to partly mediate its relationship with cognitive decline [120]. Future research should adopt multimodal approaches that integrate neuroimaging techniques (e.g., MRI and functional MRI (fMRI)) [12,121], electrophysiological measures (e.g., cortical auditory evoked potentials (CAEP)) [122], and genetic marker profiling [123], to better understand the neural and molecular mechanisms underlying cognitive change and HL. By integrating multimodal data, computational approaches can help clarify the complex bidirectional interaction (top–down and bottom–up) that mediates cognition and HL. Computational audiology, by promoting large and complex multimodal data, offers the potential to improve diagnostics, customize interventions, and monitor treatment efficacy, thereby strengthening clinical decision-making. Such efforts require a strong, collaborative commitment from the audiological, clinical, and research communities to establish consensus guidelines and standardized methods, ultimately supporting effective prevention programs and personalized intervention strategies tailored to individual needs.

### 4.9. Clinical Recommendations

Based on the findings of this scoping review, the following recommendations summarize key strategies to guide the use of cognitive evaluations, promote longitudinal monitoring, and support multidisciplinary interventions. These guidelines aim to assist audiologists, clinicians, and researchers in optimizing both auditory and cognitive outcomes.

Apply adapted cognitive assessments: To minimize auditory bias, use cognitive tools adapted or validated for individuals with HL. For example: RBANS-H (Repeatable Battery for the Assessment of Neuropsychological Status- Hearing Impaired, MoCA-HI (MoCA for Hearing Impaired) [124], HI-ACE-III (Addenbrooke’s Cognitive Examination with auditory-free instructions) [125], and CANTAB (Cambridge Neuropsychological Test Automated Battery).Promote longitudinal follow-up: Monitor auditory and cognitive outcomes beyond the first year post-implantation to assess the durability of cognitive benefits and detect late-emerging changes.Encourage multidisciplinary/interdisciplinary programs: Teams including audiologists, neuropsychologists, engineers, nurses, researchers, and clinicians are essential in providing holistic care and integrated treatment planning.Develop and implement prevention programs: Identifying and addressing modifiable risk factors, such as vascular dysfunction, social isolation, and sensory decline, may help reduce the risk of dementia and improve QoL.Increase sample sizes and research collaboration: Multicenter studies with larger and more representative samples are needed to strengthen the reliability and generalizability of findings.Address cognitive impairment in candidate selection and rehabilitation: Given the high prevalence of cognitive impairment in older CI candidates, use adapted tests and rehabilitation tools to support personalized outcomes in these patients.Include auditory-cognitive training interventions: Incorporate auditory-cognitive training into rehabilitation programs to promote neuroplasticity and cognitive recovery.Set realistic expectations: Counsel patients and families about the expected auditory and cognitive benefits of CI, provide practical strategies for daily functioning, and emphasize that improvements in auditory input may help maintain or enhance cognitive abilities and personal autonomy over time.Screening for mood disorders: HL aggravates depression, anxiety, and loneliness, which may improve after CI rehabilitation. Mood disorders can impact both auditory and cognitive outcomes. Regular screening, psychoeducation, and mental health support can help prevent the negative interaction between mood disorders and HL.Use multimodal evaluation approaches: Combine behavioral, electrophysiological (e.g., CAEP), neuroimaging (MRI/fMRI), and genetic data to better understand the neural mechanisms relating HL and cognition.Adopt computational audiology tools: Employ artificial intelligence and data-driven methods to integrate multimodal datasets, and identify patients at risk of cognitive decline based on their hearing profile and other clinical variables.

## 5. Conclusions

This scoping review comprehensively examines the latest evidence on the effects of cochlear implantation on cognitive outcomes and its relationship with speech comprehension and QoL, analyzing both longitudinal and cross-sectional studies published over the past ten years. It aims to provide a structured guide to assist clinicians and researchers in navigating the complex and sometimes inconsistent literature. Beyond summarizing outcomes, this paper highlights remaining challenges and outlines practical recommendations.

As reported, longitudinal studies demonstrated that CI leads to improvements in several cognitive domains. Executive function is the domain most frequently studied, with gains reported in 25 out of 54 total evaluations (46%). Memory significantly improved in 14 out of 23 assessments (61%) and global cognition in 13 out of the 23 test measures (57%). Effects on attention (9/16 assessments, 56%) and language (7/18 assessments, 39%) were less consistent, while visuospatial abilities improved only in 6 of 18 assessments (33%). In long-term data, 12 studies reported sustained gains in cognition from two to seven years post-implantation; the findings remain mixed, and current evidence suggests that cognitive benefits may stabilize after two years of implantation.

Studies frequently report differences in cognitive status between CI users and other hearing subgroups across multiple domains. Global cognition differed in 4 out of 7 evaluations (57%), memory in 6 out of 10 (60%), executive functions in 14 out of 31 (45%), attention in 7 out of 12 (57%), language in 2 out of 8 (25%), visuospatial abilities in 1 out of 6 (16%), and reading skills, in 2 out of 2 assessments (100%). These data support evidence that CI may facilitate recovery in higher-order cognitive domains affected by sensory deprivation, although CI users do not consistently achieve the cognitive performance levels of NH peers.

Several studies report that greater gains in speech perception are linked to improvements across cognitive domains, although the strength and consistency of these associations vary in the literature. Significant correlations between speech intelligibility and cognitive performance were observed in only 40 out of 197 cases (20%). By domain, correlations were found for global cognition 6/24 (25%), executive function 17/78 (22%), attention 4/19 (21%), language 4/24 (17%), visuospatial skills 4/20 (20%), memory 3/25 (12%), and reading abilities 2/7 (29%). These findings suggest a bidirectional relationship, in which improvements in auditory function may support cognitive recovery, while higher cognitive capacities may facilitate better speech intelligibility. Nonetheless, the limited number and generally weak strength of correlations indicate that further investigation is still needed to clarify and confirm these associations.

It is well established that hearing loss strongly impacts QoL and the psychological well-being of patients. In this review, associations between cognition and these outcomes were examined in 118 studies, with significant correlations reported only in 14 cases (12%). Despite the limited association found with cognition, improvements in QoL and well-being following cochlear implantation are frequently observed, suggesting that CI not only restores auditory function but also enhances social engagement and promotes greater daily-life independence.

The variability in outcomes likely reflects differences in study design, audiological and cognitive materials, follow-up durations, sample sizes, statistical data analysis, and variable criteria for defining cognitive normality. Despite these limitations, CI appears to alleviate the effects of sensory deprivation and support cognitive recovery, although its role in preventing or slowing dementia remains unclear.

Future research should prioritize long-term longitudinal studies and adopt multimodal approaches, integrating neuroimaging, electrophysiology, genetic data, and computational audiology to better understand auditory-cognitive interactions and develop preventive and personalized rehabilitation programs. Clinically, early identification of individuals at risk, ongoing monitoring, and tailored CI interventions are crucial for optimizing hearing, cognitive, and QoL outcomes.

## Figures and Tables

**Figure 1 jcm-14-07628-f001:**
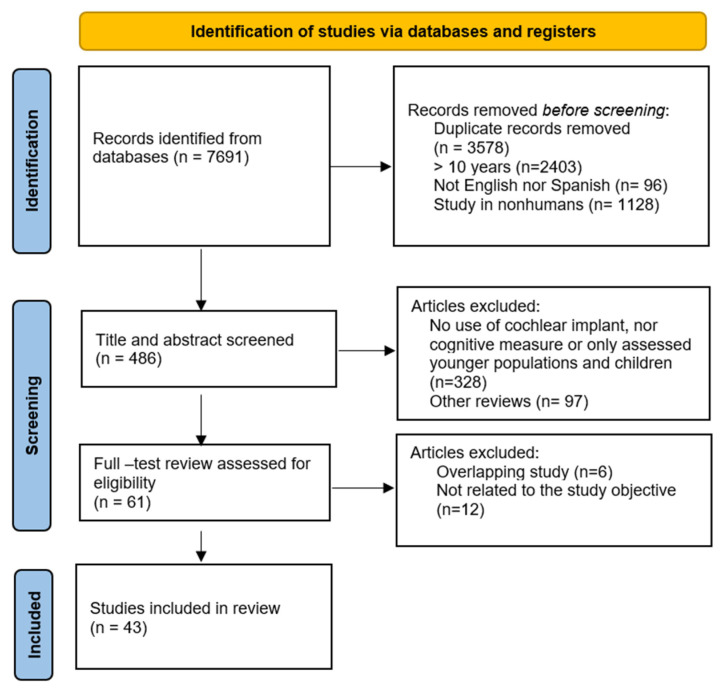
PRISMA flow diagram.

## Data Availability

No original data were generated or analyzed. Appendix A provide the raw data of correlations extracted from the reviewed studies (Longitudinal_correlations.xlsx and CrossSectional_correlations.xlsx). Cells are coded as 1 for significant correlations, 0 for non-significant correlations, and N/A for correlations that were not reported.

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
