# Peer review of "Neuropsychological Assessments to Explore the Cognitive Impact of Cochlear Implants: A Scoping Review"

_jcm, 2025, doi:10.3390/jcm14217628_

Round 1
Reviewer 1 Report
Comments and Suggestions for Authors
- The PRISMA checklist indicates no registration (e.g., PROSPERO) . Please add a brief justification of why a protocol was not registered and clarify whether a protocol document existed internally.
- This choice of appraisal tool is scientifically inappropriate for the evidence level being synthesized. The quasi-experimental designs reviewed—which compare outcomes before and after intervention (pre-post designs) or compare non-randomized groups (CI recipients vs. NH/HA controls)—represent Level 2 or 3 evidence. A checklist designed for Level 4 evidence (case series) lacks the necessary domains to rigorously assess the most critical biases inherent in this literature. Therefore, you must perform a post-hoc risk of bias assessment using a methodology designed for non-randomized interventional studies, such as the Risk of Bias in Non-randomized Studies—of Interventions (ROBINS-I) tool, or, as a minimum alternative, the Newcastle-Ottawa Scale (NOS). The revised assessment must accurately reflect the known methodological limitations of the CI literature and revise Supplementary Table 3 accordingly.
- The reported search end date of "August 2025" must be immediately corrected to the actual date the literature search was executed, thereby resolving a critical reporting anomaly.
- Explicitly incorporate a nuanced discussion in the Methodology and Discussion sections regarding the limitations of aggregating results (e.g., the 65% improvement rate) derived solely from the statistical significance (Yes/No) reported across 48 diverse tests, rather than standardizing outcomes using reported effect sizes. This clarifies the limitations of the qualitative synthesis method.
- Reframe the discussion around Executive Function. While EF shows variable gains (43%), its function as the most consistent predictor (40% correlation) of speech performance should be emphasized. This positions EF not merely as a beneficiary of reduced cognitive load, but as the primary functional bottleneck and cognitive determinant of successful CI use, particularly under challenging listening conditions (ELU model).
- Strengthen the discussion linking the profound lack of standardization in cognitive testing (48 different tests and variable cut-offs) directly to the risk of systematic assessment bias and overdiagnosis (Harbinger Hypothesis). Use this point to transition directly into the highly relevant clinical recommendation for using auditory-adapted tests (RBANS-H, MoCA-HI, ACE-III).
- Ensure the discussion section thoroughly contextualizes the findings against highly recent and similar high-impact systematic reviews, such as An et al. (2023) and Amini et al. (2023), clearly articulating how the current manuscript’s unique focus on psychological factors and QoL expands the knowledge base beyond their scope.
minor revisions are needed:
-
Terminology Consistency: Ensure consistent use of QoL metrics; for example, the abbreviation "Qo" should be uniformly rendered as "QoL".
-
Grammatical Flow: Isolated instances of run-on sentences or minor grammatical breaks need correction (e.g., refining sentences such as "noisy conditions.These findings highlights" ).
-
Data Consistency Check: A meticulous verification of the links between the quantitative percentages reported in the abstract/conclusion and the underlying data summarized in Supplementary Tables 1 and 2 is required to ensure numerical consistency across the document.
Reviewer 2 Report
Comments and Suggestions for Authors
The authors of this manuscript address a timely and clinically significant topic: the potential association between cochlear implant use and a reduced incidence of dementia. The underlying premise of the study is sound, and the research question is of great interest to the fields of audiology, neurology, and geriatrics.
However, while the topic is relevant, the execution of the study does not adhere to the fundamental guidelines required for the proposed study design. The manuscript presents itself as a systematic review, but it suffers from critical methodological flaws that undermine the validity of its findings and conclusions.
-
Lack of a Registered Protocol: A fundamental and non-negotiable standard for modern systematic reviews is the a priori registration of a study protocol (e.g., in PROSPERO). The absence of a registered protocol in this manuscript is a critical omission. This step is essential for ensuring transparency, minimizing bias, and preventing outcome switching. Risk-of-bias tools used within systematic reviews often assess whether primary studies have published protocols; for a synthesis of evidence to lack this same standard is a disqualifying flaw.
-
Deficiencies in Core Methodological Design: The study design suffers from several major weaknesses that compromise the integrity of the review. Specifically:
-
Unfocused Outcomes: The authors failed to define a single, clear primary outcome. This has resulted in an overly broad and unfocused research scope, making a rigorous synthesis challenging.
-
Ineffective Eligibility Criteria: The inclusion and exclusion criteria are not sufficiently stringent or well-defined to produce a homogenous set of studies suitable for a meaningful synthesis.
-
Arbitrary Search Parameters: The justification for the search's time horizon appears arbitrary and is not clearly explained, which may have introduced selection bias.
-
-
Discrepancy Between Stated and Actual Review Type: While the manuscript is presented as a systematic review, its execution, data extraction, and synthesis are far more characteristic of a scoping review. The authors identified primary studies with measurable, quantitative outcomes (e.g., scale-based scores, frequencies) that were amenable to statistical synthesis. However, no meta-analysis was performed, nor was a plausible justification provided for its absence. The data extraction tables and narrative summary focus on qualitative findings, which does not align with the quantitative potential of the source material and is not in keeping with the standards of a systematic review.
Despite these significant methodological issues, it is important to commend the authors on the high quality of their writing and the care taken in the narrative description of the findings from the included studies. The manuscript is well-written and the textual quality is excellent.
While the research topic is of high importance, the fundamental methodological flaws present in this manuscript are too significant to be addressed through revision. The lack of a registered protocol, coupled with a design that does not align with the core principles of a systematic review, unfortunately disqualifies the work from being considered a valid synthesis of evidence.
